# Pollutants Sorbent Made of Cotton Fabric Modified with Chitosan-Glutaraldehyde and Zinc Oxide Particles

**DOI:** 10.3390/ma14123242

**Published:** 2021-06-11

**Authors:** Vesislava Toteva, Desislava Staneva, Ivo Grabchev

**Affiliations:** 1Department of Organic Synthesis and Fuels, University of Chemical Technology and Metallurgy, 1756 Sofia, Bulgaria; vesislava@uctm.edu; 2Department of Textile and Leather, University of Chemical Technology and Metallurgy, 1756 Sofia, Bulgaria; grabcheva@mail.bg; 3Faculty of Medicine, Sofia University “St. K. Ohridski”, 1407 Sofia, Bulgaria

**Keywords:** composite materials, chitosan, zinc oxide, cotton fabric, sorbent, spills of oil, reactive dye

## Abstract

The paper reports on the preparation of composite materials by modifying cotton fabric with a layer of crosslinked glutaraldehyde chitosan containing zinc oxide particles. The ability of chitosan to form complexes with zinc ions has been used to control the size, structure, and distribution of the particles on the fiber surface. The three different obtained materials have been characterized by optical and scanning electron microscopy, Fourier-transform infrared spectroscopy (FTIR), and fluorescent analysis. It has been found that the interaction of the ZnO particles with the functional groups of chitosan affects its swelling ability in water and thus determines its sorption properties. The capacity of the materials to wipe water-soluble (textile reactive dye) and water-insoluble (crude oil and oil products) contaminants has been compared. The effect that the amount of zinc oxide has on the ability of the materials to remove contaminants has also been studied. The possibility for adsorption–desorption of the crude oil and reuse of the sorbent material has been investigated as well.

## 1. Introduction

Water pollutants can be of different origins and concentrations. Some of them occur naturally, but others arise as a result of the daily activities and mankind’s way of life. Very often, water treatment is more complicated and has to be a multi-stage process because of the presence of several contaminants [1], which can be divided into four groups (inorganic, organic, biological, and radiological). Organic substances may be crude oil, synthetic dyes, insecticides and herbicides, detergents, disinfecting cleaners, and prescription drugs [2]. The similarity between synthetic dyes and crude oil as pollutants is that even a very small amount make water aesthetically unattractive. Both are dangerous for humans and ecosystems. The difference between them is that most dyes are well dissolved in water, while crude oil and oil products are water insoluble and float on the water’s surface. Using sorbents has become one of the more effective methods for their removal from water. In this way, contaminants (dyes, crude oil, etc.) are ‘wiped’ by a material and removed from the medium. Afterward, the sorbent may be regenerated using an appropriate method, and then reused [3]. The other requirements that such a sorbent should meet are capacity for contaminant removal by sorption, a high sorption rate, and easy separation from the treated water [4]. The materials can act as adsorbents or absorbents, and via a mixed mechanism. Adsorption involves the adherence of contaminants onto a solid-phase surface, while absorption relies on capillary attraction [5].

Thus far, various inorganic, organic, natural and synthetic products have been studied and applied as sorbents. Natural products attract interest for their biodegradability, environmental friendliness, and economic efficiency [6]. These may include cotton textiles that are widely used and preferred by customers due to their desirable characteristics. Therefore, the constantly increasing amount of textile waste can be used for producing sorbents. However, to improve performance some modifications are necessary. In recent years, interest in the creation of new materials has been focused on so-called organic–inorganic nanocomposite structures, combining natural polymers with metals or metal oxides [7].

Recently, we developed a novel sorbent material based on natural products, such as cotton fabric modified with glutaraldehyde-crosslinked chitosan, and obtained zinc oxide particles in situ [8]. These composite materials have good sorption properties for crude oil and oil products. Their advantages are their effectiveness and ease of use, as well as possible reusability. The disadvantage of utilizing them is the multi-stage treatment of the cotton fabric with chitosan, zinc ions, and glutaraldehyde. It has also been found that the greater the thickness of the sorbent layer in the composite material and the presence of a larger amount of ZnO particles in it, the lower the sorption capacity and effectiveness are.

It is worth elucidating the role of each component of the composite material (cotton fabric, crosslinked chitosan layer, and ZnO particles) in the sorption process of various pollutants. The cotton fabric is hydrophilic but has a negative zeta potential. Hence, the electrostatic interaction of the surface of a textile with the solutes in an aqueous solution plays an important role in adsorption and desorption kinetics [9]. Consequently, because of this interaction, the sorption properties of a material without further treatment are limited to being applied to both water-soluble and water-insoluble pollutants.

It is known that cotton modification with chitosan increases the absorption of a reactive dye [10]. Li et al. found that chitosan-coated filter paper has excellent super hydrophilicity and underwater superoleophobicity. These properties are the reason for the exceptional results in the separation of an oil-in-water emulsion [11]. On the other hand, the nanocomposite chitosan hydrogel/ZnO shows a higher swelling capacity compared to pure chitosan hydrogel and this depends on the ZnO particle content. [12]. The great number of hydroxyl groups in cotton cellulose or hydroxyl and amino groups in chitosan can form a polymer–metal complex with metal ions, such as zinc, copper, silver, etc. [13,14]. Thus, chitosan can stabilize and control the structure of the particles obtained by the action of sodium hydroxide and heating [15]. It is expected that the sorption process of the composite materials can be facilitated by the characteristic morphology of ZnO particles that tend to form flower-like structures [16].

The aim of this study is to prepare composite materials from modified cotton fabric with glutaraldehyde-crosslinked chitosan and zinc oxide particles using a procedure that is more facile than those previously applied [8]. The structures and new characteristics of the newly prepared materials have been subjected to optical and scanning electron microscopy, FTIR, and fluorescent analysis. The sorption properties of the materials regarding textile reactive dye and against crude oil and oil products are compared. The effects that the amount zinc ions and their conversion into zinc oxide particles have on the ability of the materials to remove contaminants has been evaluated. The possibility for adsorption–desorption cycles of oil and the regeneration and reuse of the sorbent material have also been investigated.

## 2. Materials and Methods

### 2.1. Materials

A bleached and unmercerized, plain-woven, 100% cotton fabric (Ct), with a surface weight of 145 ± 5 g/m^2^ was used throughout the work. Zn(NO_3_)_2_·6H_2_O (CAS Number:10196-18-6, from Sigma-Aldrich (Darmstadt, Germany) and NaOH were purchased from from Sigma-Aldrich (Darmstadt, Germany. Glacial acetic acid (CAS Number: 64-19-7, from Sigma-Aldrich (Darmstadt, Germany) and glutaraldehyde (25% aqueous solution) (CAS Number: 111-30-8) were used without further purification and were obtained from Sigma-Aldrich (Darmstadt, Germany. Chitosan (with a molecular weight ranging from 600,000 to 800,000 was purchased from Acros Organics, Geel, Belgium. All solutions were made with distilled water.

### 2.2. Reactive Dye Drimarene Rot K-7B and Crude Oil and Oil Products

The reactive dye Drimaren Red K-7B (Clariant, Basel, Switzerland) was used for the preparation of the model dye solution with a concentration of 0.04 g L^−1^. Three possible contaminants (crude oil, diesel fuel, and motor oil-15W/40) were used to treat wastewater in an imitation of an oil spill. The quantity of the oil and other product to water was a ratio of 1:50. The main physical characteristics of tested oil samples are presented in Table 1.

### 2.3. Preparation of Composite Materials

#### 2.3.1. The Preparation of Different Solution

Three different treatment solutions were prepared as follow:1)The chitosan (2.7% *w*/*v*) was dissolved in water under stirring, with the gradual addition of glacial acetic acid (1% *v*/*v*) to obtain a clear viscous solution;2)The chitosan (2.7% *w*/*v*) and Zn(NO_3_)_2_x6H_2_O (2.7% *w*/*v*) were dissolved in water with the gradual addition of glacial acetic acid (1% *v*/*v*) to obtain a clear viscous solution;3)Zn(NO_3_)_2_x6H_2_O was dissolved in water. The quantity of Zn^2+^ was 20% owf (on weight of fabric).

#### 2.3.2. The Fabric Treatment

Three different samples were obtained and named as Ch, ChZn, and ZnChZn.

1)Sample Ch

The cotton fabric was impregnated with solution (1) containing chitosan and dried at 80 °C for 10 min. The volume of the solution was of a liquor to goods ratio of 4:1. Next, the sample was impregnated with a glutaraldehyde water solution (5% *w*/*w* to chitosan) with a liquor to goods ratio of 3:1 and dried again (80 °C for 10 min), then it was thoroughly washed with deionized water and finally dried at room temperature for 24 h.

2)Sample ChZn

The cotton fabric was impregnated with solution (2) containing chitosan and Zn(NO_3_)_2_, then dried at 80 °C for 10 min. The solution volume was of a liquor to goods ratio of 4:1. Next, the sample was impregnated with glutaraldehyde water solution (5% w/w to chitosan) with a liquor to goods ratio of 3:1 and dried again (80 °C for 10 min). Finally, the sample was soaked in sodium hydroxide solution with a ten-time stoichiometric excess to zinc ions and treated thermally at 80 °C for 30 min. The obtained composite materials were washed with water and allowed to dry at a room temperature.

3)Sample ZnChZn

The cotton fabric was impregnated with solution (3) containing Zn^2+^, then dried at 80 °C for 10 min. The solution volume was of a liquor to goods ratio of 2:1. Then the procedure was applied to obtain sample ChZn. The possible reaction between the reagents is presented in Figure 1.

### 2.4. Analysis

The pristine cotton fabric and the composite materials, Ch, ChZn and ZnChZn, in the dry and wet states were analyzed using an optical microscope (Levenhuk Rainbow D50L PLUS 2M Digital Microscope, Moonstone c 2 Mpx Digital Camera, PRC, controlled by Levenhuk, Inc. (USA)). The magnification of the pictures was ×40.

The surface morphology of the composite materials and the formation of ZnO particles were analyzed using a scanning electron) microscope (SEM) JSM-5510 (Jeol Ltd., Tokyo, Japan) operated at a 10-kV acceleration voltage. The investigated samples were coated with gold by a JFC-1200 fine coater (Jeol Ltd., Tokyo, Japan) before imaging.

The photoluminescence (PL) spectra of the composite material were obtained by a VARIAN Cary Eclipse model spectrometer, Darmstadt, Germany at room temperature with resolution of 5 nm.

IR analysis was carried out on an Infrared Fourier Transform spectrometer (IRAffinity-1, Shimadzu, Tokio, Japan) equipped with a diffuse-reflectance attachment (MIRacle Attenuated Total Reflectance Attachment). Measurements were done using a spectral range of 600–4000 cm^−1^.

X-ray fluorescence (XRF) spectrometry was used to define the percentage of ZnO in composite materials Ch and ChZn. A wavelength dispersive (WDXRF) technique was used and Rigaku Supermini200 (Bruker, Hanau, Germany) equipment was used. Data were processed by EZScan software (Bruker, Hanau, Germany). X-ray tube: 50 kV, 200 W Pd-anode, holder 52 mm round holder/samples size 30 mm diameter (surface irradiated).

### 2.5. Sorbent Properties of the Composite Materials

#### 2.5.1. The Sorption of Crude Oil and Oil Products

The oil sorption capacity *S* (*g*/*g*) of the composite material to crude oil and oil products was calculated from the mass of material before (*M*_1_) and after (*M*_2_) sorption using the following equation:(1)$$Sorption\;capacity\;\left(g/g\right)=\frac{M2-M1}{M1}$$

The experiments were performed using samples (0.05 g), which were dipped for 10 min into mixtures of 50 mL of water and 1 mL of oil.

#### 2.5.2. Reusability of ChZn Composite Material

The composite (ChZn) regeneration and its repeated use for oil sorption were tested using the following experiment. Samples with different weights (0.1 g and 0.2 g) were put into 50 mL water, containing 1 mL of oil for 3 min. The material was then taken out from the water and drained for one minute. The sample was weighed and its sorption capacity was determined using Equation (1). The regeneration was done by immersion of the sample in n-hexane for 3 min.

#### 2.5.3. Discoloration of the Reactive Dye Drimaren Rot K-7B Water Solution

Weighed amounts of Ch, ChZn, or ZnChZn composite materials (80 mg) were immersed in 20 mL of Drimaren Red K-7B aqueous solution (0.04 g L^−1^) at room temperature. A total of 3 mL of solution was periodically taken, measured spectrophotometrically, and placed back in the same beaker so that the volume of the liquid was kept constant.

The absorbance of the dye at the wavelength of its maximum absorbance (at 550 nm) was used to calculate the percentage of discoloration efficiency using Equation (2):(2)$$Discoloration\;efficiency(\%)=\left[\frac{A0-At}{A0}\right]\times 100$$
where *A*_0_ is the initial absorbance and *A_t_* is the absorbance at time *t*.

## 3. Results and Discussion

### Morphological Properties of Composites

The obtained composite materials were composed of biodegradable renewable biopolymers, such as cotton and chitosan. To improve their sorption properties, inorganic zinc oxide, obtained in situ, was added to the organic materials, which provided control of its distribution, size and shape. Glutaraldehyde was used as a crosslinker of the chitosan layer.

Figure 2 presents photographs taken with a digital camera under optical microscope observation of the composite materials Ch, ChZn and ZnChZn and cotton fabric. The comparison of Figure 2(1d) and Figure 2(2d) shows that the individual yarn fibers differ notably after the application of a chitosan crosslinked with glutaraldehyde. The shape and amount of voids in the fabric structure are also well visible. The application of the solution containing chitosan and Zn(NO_3_)_2_, the crosslinking with glutaraldehyde, and the synthesis of zinc oxide particles led to the bonding of the individual fibers and the filling of the gaps (Figure 2(3d)). The multilayer coating with the first impregnation with Zn(NO_3_)_2_ solution and subsequent treatment, as in the ChZn sample, led to the formation of a uniform layer on the fabric surface (Figure 2(4d)).

After irrigation, the hydrophilic cotton fabric retained water in the interfiber spaces, but the fabric voids were visible (Figure 2(1w)). Wetting of the Ch, ChZn, and ZnChZn samples made the crosslinked chitosan swell. The gaps between the woven threads decreased and almost disappeared in sample ZnChZn.

Figure 3 shows micrographs of the cotton fabric and of both composite materials Ch and ChZn, at different magnifications. As can be seen, the chitosan-coated material retained the structure of the fabric. The plain weave of the fabric and the individual fibers forming the yarn were quite visible, but they were glued together by a thin layer of chitosan. The surface of the ChZn fabric was covered with a film that had defects and cracks in various places. Evenly distributed small white grains of zinc oxide were included throughout the structure of the film. At a magnification of ×1500, it can be seen that the chitosan adhered the individual fibers with a thin film, in some places preserving the characteristic microgaps of the fabric structure. In composite material ChZn, the individual fibers are almost indistinguishable, as they are covered with a layer with a granular structure of white spherical particles, distributed relatively evenly on the surface.

Figure 4 shows micrographs of the ChZn composite material at a higher magnification. Spherical zinc oxide particles are composed of plates of nanoscale-thickness that form a flower-like structure with a diameter about 1–2 μm. Some particles are covered by the chitosan layer, while others are on its surface.


*FTIR Characterization of Composite Materials*


The FTIR spectra of the unmodified cotton fabric and of composite materials Ch, ChZn, and ZnChZn were compared and analyzed (spectra not shown). Figure 5 presents the spectra of Ch, ChZn, and ZnChZn as a result of subtracting a spectrum of the cotton fabric for better visualization of the new characteristic peaks that appeared as a result of the modification. The presented region of 2000 cm^−1^ to 450 cm^−1^ contains peaks characterizing the interaction of ZnO particles with chitosan. The presence of a peak at 1592 cm^−1^ could be attributed to the –NH group and its intensity increased in samples ChZn and ZnChZn in the presence of ZnO particles. This band is not observed in the spectrum of Ch because the chitosan NH_2_ groups and glutaraldehyde participate in a crosslinking reaction and this generated Schiff bases. The peak at 1396 cm^−1^ corresponding to –OH groups increases in sample ChZn and ZnChZn and this indicates the interaction of ZnO particles with these groups. The absorption peaks at 1084 cm^−1^ are ascribed to the C–O stretching group. The increased intensity of the peak at 874 cm^−1^ in the spectrum of ChZn confirms the stabilization of ZnO particles with chitosan [17]. The new absorption peak at 651 cm^−1^ appears due to the attachment of the amide group and the stretching mode of ZnO [18,19]. The bands at 582 cm^−1^ may be ascribed to the Zn–O bond [20].


*Photoluminescence Analysis of the Composites*


Figure 6 shows the photoluminescence spectra of samples Ch, ChZn and ZnChZn in the solid state. After crosslinking with glutaraldehyde, chitosan emits a blue–green fluorescence with a wide spectral band with maxima at 489 and 507 nm, which are due to conjugated π bonding and π–π* transitions of the azomethine fragments [21]. The presence of zinc oxide in the crosslinked chitosan film is the reason for the weakening of the fluorescence emission of chitosan and for the appearance of a new peak at 475 nm. The peak arises due to a transition between interstitial zinc and the zinc vacancy level [22]. Increasing the amount of zinc oxide in the composite material enhances this effect.

The sample compositions obtained from X-ray fluorescence (XRF) analyses are reported in Table 2. The column labeled as “matrix” shows the amount of the main components that make up the materials. These are carbon (C), nitrogen (N), hydrogen (H) and oxygen (O). The results showed that sample Ch does not contain zinc (Zn). The other presented components are in small quantities. The sample ChZn contains 10.25 mass% Zn, while the sample ZnChZn contains 12.19 mass%. This result confirmed that the impregnation of cotton fabric with Zn ion solution leads to a small increase in the Zn component in composite material ZnChZn.


*Sorption Properties to Oil and Oil Products*


Materials Ch, ChZn and ZnChZn were investigated as sorbents for crude oil or oil products spills. The experiments were performed using samples (0.05 g) that were put into mixtures of 50 mL of water and 1 mL of oil for 10 min. The weight of sorbent was chosen in agreement with our previous study [8]. According to this, samples with less weight and a smaller area “moistened” more evenly for this period of time. In larger samples, a part of the surface fails to come into contact with the oil, and thus their sorption capacity is smaller.

Figure 7 shows that the largest sorption capacity for oil was with sample ChZn, modified with crosslinked chitosan and zinc oxide, followed by sample Ch, treated only with crosslinked chitosan. Slightly lower results were obtained for sample ZnChZn, containing a higher amount of zinc oxide. The sorption properties of composite material ChZn were tested for wiping motor oil or diesel fuel and the results were compared to those obtained for oil.

Figure 8 shows that the sorption capacity of ChZn for oil is almost twice as high as that for diesel fuel and approximately twice as low as for motor oil. The decisive factors in the case are, on the one hand, the different compositions and viscosities of crude oil and its products, and, on the other hand, the structure and oleophilic properties of materials. Motor oil contains hydrocarbons with a chain length longer than that of diesel fuel. Crude oil consists of a full range of different hydrocarbons with various molecular weights and other organic compounds. The tested motor oil (15W/40) has a viscosity over 30 times higher than that of crude oil and diesel fuel and contains a package of additives for its maintenance under winter conditions. The high viscosity and long chain hydrocarbons of motor oil can cause two opposite effects: difficult penetration into the pores of the composite material and improved sorption by adhesion to the surface of the material [21,23]. For this short experimental time of 10 min, ChZn was effective as adsorbent for more viscose oil products, and the interaction was based on hydrophobic and van der Waals forces.

The performance of the obtained materials was determined by their ease of use, stability, and good and fast sorption ability. Their economic efficiencies depend on the possibility of their easy regeneration and reuse. Therefore, it was interesting to study the possibility of repeated regeneration and application of the composite material as a sorbent until the complete elimination of the oil spill.

Figure 9 compares the sorption capacity of samples from the ChZn material of different weights (0.1 g and 0.2 g) to the complete removal of the oil spill.

Each sample was placed for 3 min in mixtures of 50 mL water and 1 mL oil, then drained for 1 min and immersed in hexane again for 3 min. The regenerated sample was placed again in the oil-in-water mixture and the above mention procedure was repeated. The experiment was carried out until the oil was completely separated or until the initial weight of the water was reached (*n* times). As Figure 9 shows, the heavier sample completely sorbs the oil by three immersions, although it shows a lower sorption capacity separately, while the removal of the oil with the smaller sample requires five sorption–desorption cycles.

Sorption Properties to Dissolved in Water Reactive Dye Drimarene Rot K-7B

The sorbent capacity against Drimarene K-7B dye dissolved in water of the composite materials and of unmodified cotton fabric were compared. According to Figure 10, the discolorization of the dye solution was fastest with sample Ch (27%), following by sample ZnChZn (18%) at 140 min. The slowest process was in the presence of sample ChZn (9%). An interesting, but expected result was observed in the case of immersing pristine cotton fabric into the dye solution. As a result, the color intensity increased by about 3% for 140 min. It is known that in aqueous solution reactive dye dissociates into colored anions and colorless cations. The cotton fabric is a hydrophilic material but has a negative zeta surface potential. This was the reason for the electrostatic repulsion between its surface and the negative dye ions. As a result, the adsorption process was hindered, but the cotton fabric swelled and absorbed a certain amount of water, which led to an observed change in the intensity of the solution. Tissue modification with chitosan introduces new amino groups to the surface. Their partial protonation in distilled water changed the surface charge of fabrics and facilitated dye adsorption. The presence of the ZnO particles led to increasing the hydrophobicity of the fabric. As the SEM images show, the particles had a flower-like structure, capable of adsorbing more dye molecules. The quantity of the particles was higher in sample ZnChZn. Spherical ZnO nanoparticles were proved to be effective adsorbents of dyes due to their highly developed surface areas, rich surface groups with high reactivity, and adsorption capacity [24]. Therefore, two reasons affected the rate of discoloration of the dye solution. One was the presence of chitosan film and its ability to swell, and the other was the zinc oxide particles on the surface of the cotton fibers and their adsorption capacity. There were particles on the fiber surface in sample ChZn, but many of them were covered by a chitosan film. ZnO particles take part in the chelation of some hydroxyl and amine groups and restrict the expansion of polymer chains [12]. That may explain why the discoloration is retarded in the presence of sample ChZn.

## 4. Conclusions

Three composite materials comprising different quantities of ZnO particles have been prepared. The ability of chitosan to form complexes with zinc ions has been exploited to obtain ZnO particles with a controlled size, structure, and distribution over the fiber surface. The interaction of the ZnO particles with the chitosan functional group affects its swelling ability in water and its sorption properties. Composite material ChZn exhibits the best sorption capacity for the oil and the worst for the dissolved dye Drimarene K-7B. This material acts as an adsorbent and its sorbtion capacity is better against the more viscous motor oil, which contains long chain hydrocarbons and a package of additives. The cotton–chitosan composite material (Ch) exhausts the dissolved dye faster than other materials because it alters the surface of cotton fabric and minimizes the fabric repulsion forces to dye molecules. This improves its adsorbent properties, while the swelling of the cross-linked chitosan layer allows the dye to be also absorbed by capillary attraction [25]. The possibility for adsorption–desorption cycles of oil and regeneration and reuse of sorbent material have also been investigated. Material ChZn was successfully regenerated with hexane and reused for the full removal of the oil spill.

In this study, the newly obtained composite materials were studied only as sorbents in contaminated water. However, the presence of ZnO particles also suggests their use as effective photocatalysts capable of facilitating the generation of reactive oxygen species during photoirradiation. As visible light is 44% of all solar energy, alloying ZnO with magnesium, calcium, and iron ions can change its light absorption and improve the process of organic pollutant degradation.

## Figures and Tables

**Figure 1 materials-14-03242-f001:**
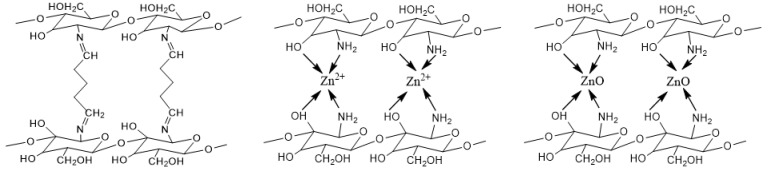
Schematic presentations of the interactions of chitosan with glutaraldehyde; chitosan with Zn ions and ZnO particles.

**Figure 2 materials-14-03242-f002:**
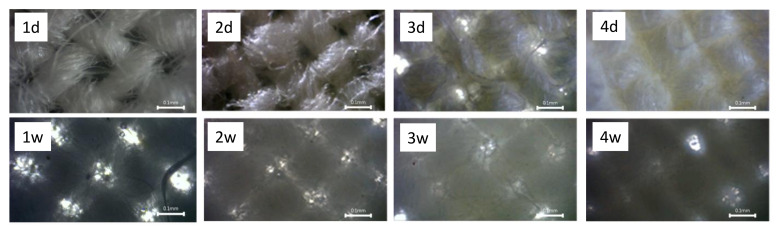
Optical microscope photographs of cotton fabric (**1d**,**1w**) and composite materials Ch (**2d**,**2w**); ChZn (**3d**,**3w**); and ZnChZn (**4d**,**4w**). In the top row, the samples are in a dry state, and, in bottom row, are after wetting with water (magnification of the pictures is ×40).

**Figure 3 materials-14-03242-f003:**
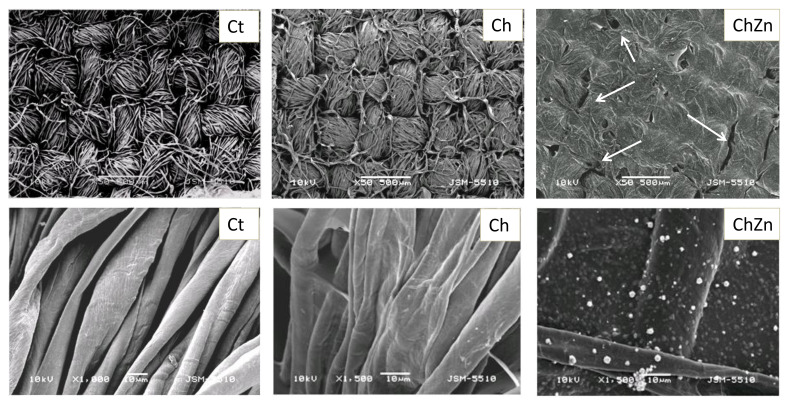
SEM images of the surface of a cotton fabric Ct and fabrics treated with Ch and ChZn at different magnifications (×50 and ×1500).

**Figure 4 materials-14-03242-f004:**
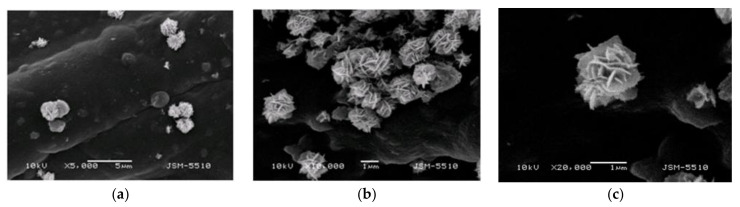
SEM images of zinc oxide particles structure and their distribution in composite material ChZn at different magnification (**a**) ×5000, (**b**) ×10,000; (**c**) ×20,000.

**Figure 5 materials-14-03242-f005:**
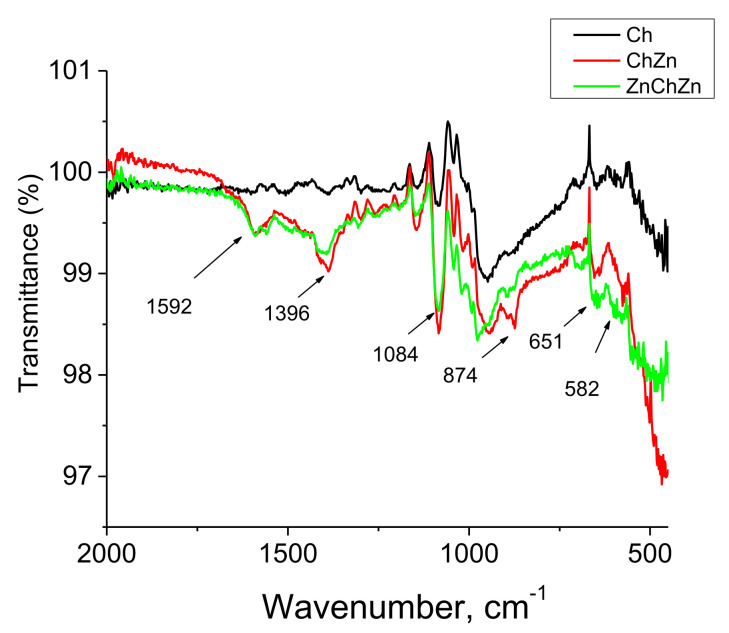
FTIR spectra of the composite materials Ch, ChZn, and ZnChZn as a result of substracting a spectrum of pristine cotton fabric.

**Figure 6 materials-14-03242-f006:**
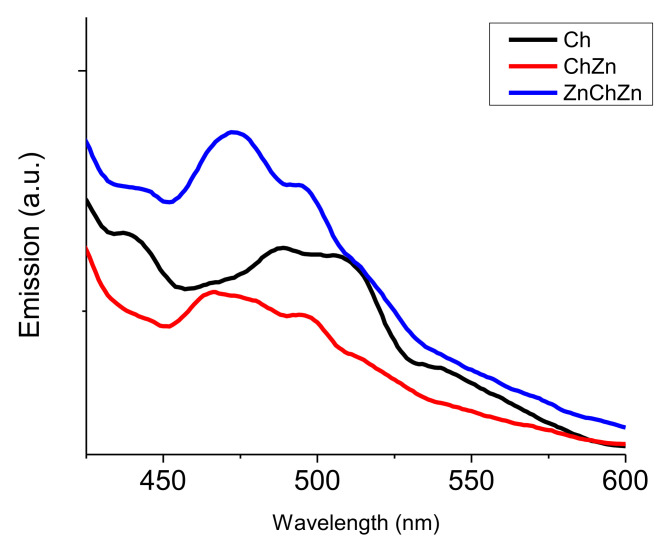
Photoluminescence spectra (λ_exc_ = 330 nm) of Ch, ChZn, and ZnChZn.

**Figure 7 materials-14-03242-f007:**
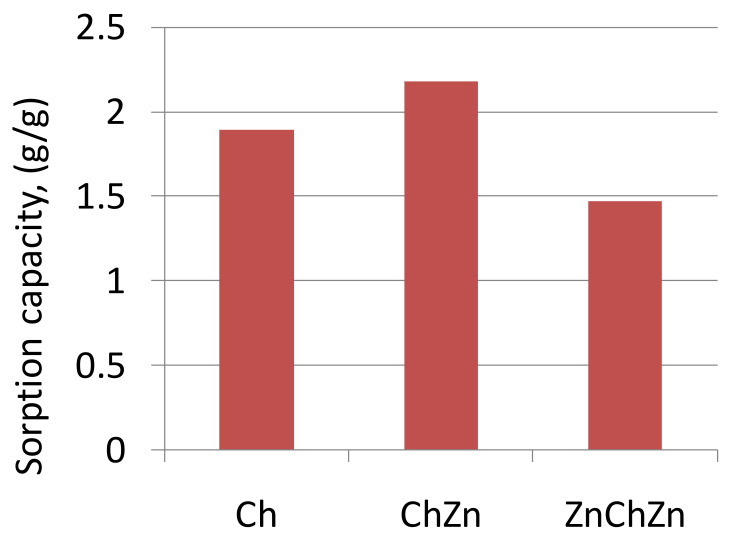
Sorption capacity of composite materials Ch, ChZn, and ZnChZn for oil.

**Figure 8 materials-14-03242-f008:**
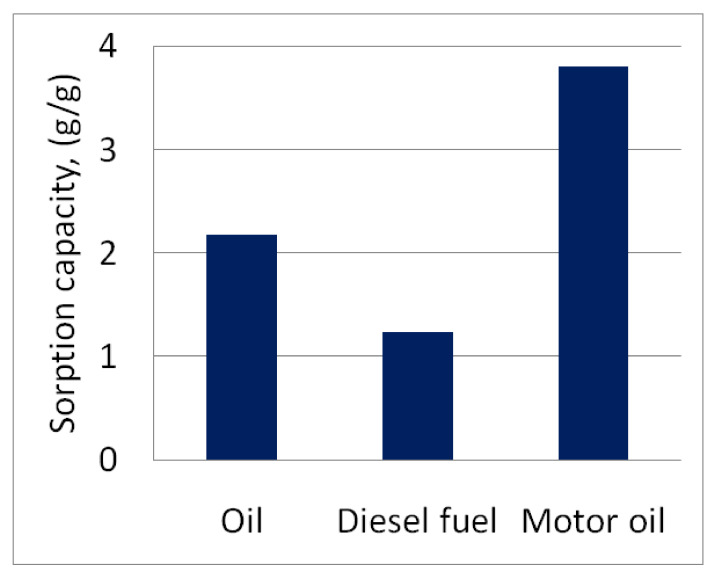
Sorption capacity of ChZn sample relative to oil and its products.

**Figure 9 materials-14-03242-f009:**
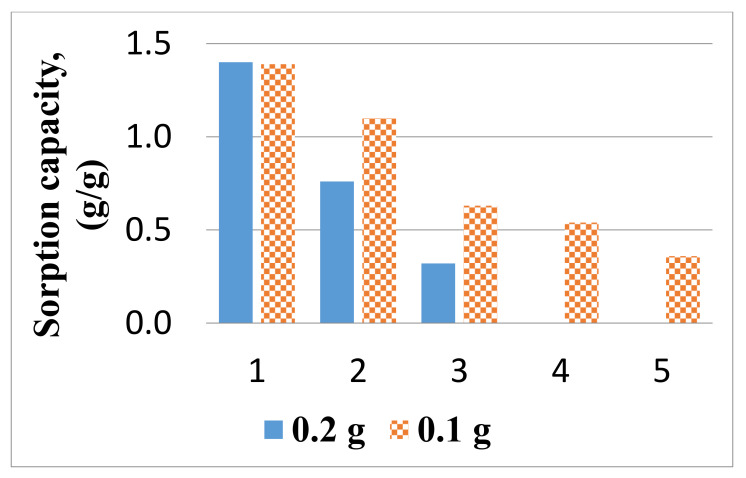
Oil sorption by sample of composite material ChZn of different weights.

**Figure 10 materials-14-03242-f010:**
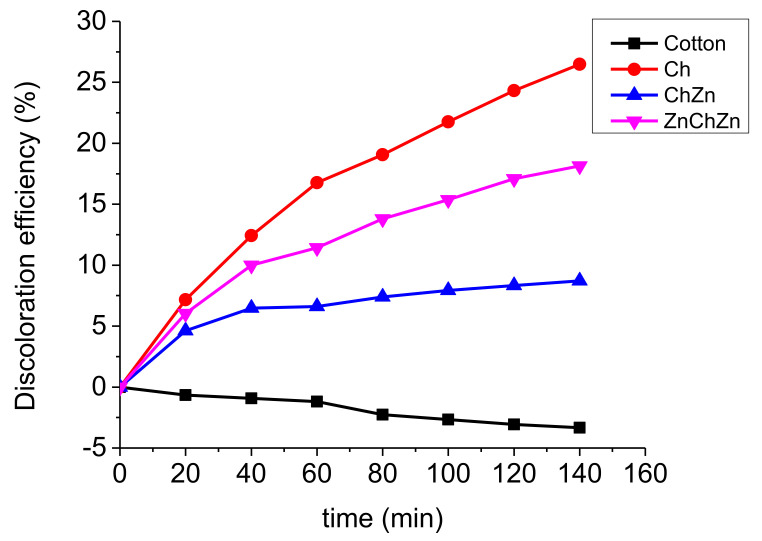
Discoloration of Drimarene K-7B dye water solution (0.04 g L^−1^) according to time by textile materials: pristine cotton, Ch, ChZn, and ZnChZn (4 g L^−1^).

**Table 1 materials-14-03242-t001:** Physical characteristics of oil samples.

Sample	Density, g/cm^3^ (15 °C)	Viscosity, mm^2^/s (40 °C)
oil	0.830	3.3
diesel fuel	0.827	2.9
motor oil 15W/40	0.940	106.0

**Table 2 materials-14-03242-t002:** Surface composition of samples Ch, ChZn, and ZnChZn with XRF analysis.

Component Mass%	Mg	Al	Si	P	S	Cl	K	Ca	Ti	Fe	Zn	Matrix
Ch	0.01	0.01	0.00	0.00	0.03	0.06	0.03	0.03	0.30	0.02	0.00	99.50
ChZn	0.67	0.25	0.27	0.08	0.18	0.33	0.54	1.39	1.46	0.14	10.25	84.44
ZnChZn	0.58	0.17	0.33	0.06	0.18	0.32	0.72	1.33	1.97	0.15	12.19	82.01

## Data Availability

Data available in a publicly accessible repository.

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
