# Peer review of "Pollutants Sorbent Made of Cotton Fabric Modified with Chitosan-Glutaraldehyde and Zinc Oxide Particles"

_materials, 2021, doi:10.3390/ma14123242_

Round 1
Reviewer 1 Report
Manuscript Number: materials-1228083
Title: Pollutants Sorbent Made of Cotton Fabric Modified with Chitosan-Glutaraldehyde and Zinc Oxide ParticlesComments:
Overview
The paper reports some interesting results. The articles can be published after major revisions. The following aspects should be addressed:
Materials and Methods
- Page 3, lines 106-115, the authors report that two different treatment solutions were prepared but they describe 3 different systems. Can the authors explain this better?
- Page 5, lines 171-173, how are the absorbance measurements conducted? recording the dye in solution before and after treatment? I suggest describing the procedure.
- Page 9, line 267, check typos in caption fig.7.
Results and discussion
- Page 9, lines 276-278 the authors write: “The high viscosity and long chain hydrocarbons of motor oil can cause two opposite effects: difficult penetration into the pores of the composite material and improved sorption by adhesion to the surface of the material.” Why the global effect gives an increase in sorption capacity of ChZn sample?
- Page 10 line 313-320. The authors write: “The presence of the ZnO particles leads to increasing the hydrophobicity of the fabric. ……..The quantity of the particles is higher in sample ZnChZn. There are particles on the fiber surface in sample ChZn, but many of them are covered with a chitosan film. ZnO particles take part in the chelation of some hydroxyl and amine groups and restrict the expansion of polymer chains. That may explain why the discoloration is retarded in the presence of sample ChZn.”
It is not well clear the discussion: chitosan alone introduces amino groups to the surface which, partially protonated, changes the surface charge of fabrics and facilitates dye adsorption. ZnO particles on the contrary increase the hydrophobicity of the fabric and reduce the dye adsorption, and this justify the decrease observed in figure 10 for samples containing ZnO. But why ZnChZn gives a higher Discoloration Efficiency in respect of ZnCh?

Author Response
1) Page 3, lines 106-115, the authors report that two different treatment solutions were prepared but they describe 3 different systems. Can the authors explain this better?
The word “two” has been changed to “three”
2) Page 5, lines 171-173, how are the absorbance measurements conducted? recording the dye in solution before and after treatment? I suggest describing the procedure.
Done
3) Page 9, line 267, check typos in caption fig.7.
The caption of fig.7 has been changed.
4) Page 9, lines 276-278 the authors write: “The high viscosity and long chain hydrocarbons of motor oil can cause two opposite effects: difficult penetration into the pores of the composite material and improved sorption by adhesion to the surface of the material.” Why the global effect gives an increase in sorption capacity of ChZn sample?
The explanation is added in the text.
5) Page 10 line 313-320. The authors write: “The presence of the ZnO particles leads to increasing the hydrophobicity of the fabric. ……..The quantity of the particles is higher in sample ZnChZn. There are particles on the fiber surface in sample ChZn, but many of them are covered with a chitosan film. ZnO particles take part in the chelation of some hydroxyl and amine groups and restrict the expansion of polymer chains. That may explain why the discoloration is retarded in the presence of sample ChZn.”
It is not well clear the discussion: chitosan alone introduces amino groups to the surface which, partially protonated, changes the surface charge of fabrics and facilitates dye adsorption. ZnO particles on the contrary increase the hydrophobicity of the fabric and reduce the dye adsorption, and this justify the decrease observed in figure 10 for samples containing ZnO. But why ZnChZn gives a higher Discoloration Efficiency in respect of ZnCh?
The explanation is added in the text.
Reviewer 2 Report
The paper by Toteva V. et al. deals with the preparation of composite materials by modifying cotton fabric with a layer of crosslinked glutaraldehyde chitosan containing zinc oxide particles. The sorption properties of this material to textile reactive dye, crude oil and oil products have been studied. The manuscript can be published after some improvements indicated below.
- English should be improved.
- Scale bars in Fig. 2 should be added.
- Figure captions should be improved. The characterization methods should be added. For instance, Fig. 3 SEM should be indicated in the figure caption.
- Line 205-206, ‘the surface of ChZn fabric is covered with a film that has defects and cracks in various places’. These observations should be indicated in the images because there are not obvious. The images in Fig. 3 should be improved to be brighter.
- The reason why the particles of zinc oxide have this structure should be indicated. Have these particles been characterized separately?
Author Response
1) English should be improved.
Improved
2) Scale bars in Fig. 2 should be added.
Done
3) Figure captions should be improved. The characterization methods should be added. For instance, Fig. 3 SEM should be indicated in the figure caption.
Done in Fig. 3 and Fig. 4
4) Line 205-206, ‘the surface of ChZn fabric is covered with a film that has defects and cracks in various places’. These observations should be indicated in the images because there are not obvious. The images in Fig. 3 should be improved to be brighter.
Done
5) - The reason why the particles of zinc oxide have this structure should be indicated. Have these particles been characterized separately?
The ZnO particles have been obtained in situ and were not characterized separately.
A text with an explanation of the advantages of spherical nanoparticles is added to the manuscript.
Reviewer 3 Report
The authors present pollutants sorbent made of cotton fabric modified with chitosan-glutaraldehyde and zinc oxide particles. The paper has potential; however, there are many flaws and shortcomings that need serious attention for publication. Hence, I would decide the manuscript's fate based on the major revision of the following comments.
1- Define all the abbreviations at the first instance of their mention in the manuscript.
2- provide the CAS numbers of all the chemicals and substrates used in the experiments.
3- provide a well pronounced, defined and highlighted scalebar in figure 2, 3 and 4.
4- How did the authors confirmed the presence of Ch, ChZn and ZnChZn in the woven materials? What was the atomic percentage of the individual elements? did the authors study these properties via EDS?
5- define y-axis in figure 5.
6- Why did the authors specifically preferred the ZnO particles over other 1D and 2D ZnO morphologies?
7- Explain the limitations to this study and the future directions in the conclusion section.
Author Response
1- Define all the abbreviations at the first instance of their mention in the manuscript.
Done
2- provide the CAS numbers of all the chemicals and substrates used in the experiments.
Done
3- provide a well pronounced, defined and highlighted scalebar in figure 2, 3 and 4.
Done
4- How did the authors confirmed the presence of Ch, ChZn and ZnChZn in the woven materials? What was the atomic percentage of the individual elements? did the authors study these properties via EDS?
X-ray fluorescence (XRF) spectrometry has been used to confirm the presence and quantity of ZnO.
The results were added to the text.
5- define y-axis in figure 5.
Done
Round 2
Reviewer 1 Report
The paper can be accepted in this form. All the suggestions have been modified.
Reviewer 3 Report
The authors have sufficiently revised the manuscript as per the revision comments. I believe the paper is now ready for publication. Hence, I would like to accept the paper publication in materials.